# Passing of Nanocarriers across the Histohematic Barriers: Current Approaches for Tumor Theranostics

**DOI:** 10.3390/nano13071140

**Published:** 2023-03-23

**Authors:** Kamil Gareev, Ruslana Tagaeva, Danila Bobkov, Natalia Yudintceva, Daria Goncharova, Stephanie E. Combs, Artem Ten, Konstantin Samochernych, Maxim Shevtsov

**Affiliations:** 1Institute of Cytology of the Russian Academy of Sciences (RAS), 194064 Saint Petersburg, Russia; 2Department of Micro and Nanoelectronics, Saint Petersburg Electrotechnical University “LETI”, 197022 Saint Petersburg, Russia; 3Personalized Medicine Centre, Almazov National Medical Research Centre, 2 Akkuratova Str., 197341 Saint Petersburg, Russia; 4Department of Radiation Oncology, Technishe Universität München (TUM), Klinikum rechts der Isar, Ismaningerstr. 22, 81675 Munich, Germany; 5Institute of Life Sciences and Biomedicine, Far Eastern Federal University, 690922 Vladivostok, Russia

**Keywords:** nanoparticles, tumor theranostics, cell-penetrating peptide, histohematic barrier, blood–tissue barrier, blood–tumor barrier

## Abstract

Over the past several decades, nanocarriers have demonstrated diagnostic and therapeutic (i.e., theranostic) potencies in translational oncology, and some agents have been further translated into clinical trials. However, the practical application of nanoparticle-based medicine in living organisms is limited by physiological barriers (blood–tissue barriers), which significantly hampers the transport of nanoparticles from the blood into the tumor tissue. This review focuses on several approaches that facilitate the translocation of nanoparticles across blood–tissue barriers (BTBs) to efficiently accumulate in the tumor. To overcome the challenge of BTBs, several methods have been proposed, including the functionalization of particle surfaces with cell-penetrating peptides (e.g., TAT, SynB1, penetratin, R8, RGD, angiopep-2), which increases the passing of particles across tissue barriers. Another promising strategy could be based either on the application of various chemical agents (e.g., efflux pump inhibitors, disruptors of tight junctions, etc.) or physical methods (e.g., magnetic field, electroporation, photoacoustic cavitation, etc.), which have been shown to further increase the permeability of barriers.

## 1. Introduction

Nanoparticle-based systems in recent decades have been employed for tumor diagnostics and therapy (i.e., theranostics) [1]. However, although numerous approaches have been followed to enhance the retention of nanocarriers in the tumor site, few have shown significant success, as the tumor harbors a vascular barrier system (blood–tumor barrier (BTB)) that limits the passage of agents from the bloodstream into the tumor parenchyma [2,3,4]. Therefore, significant efforts are devoted to the development of novel molecules with the ability to facilitate the crossing of the BTB. In addition to the existing BTB, certain organs and tissues prevent the transport of hydrophilic agents and large molecules (>70 kDa) due to the presence of blood–tissue barriers. Examples of such barriers are the blood–brain, blood–retina, blood–testis, blood–thymus, blood–placenta, and gut barriers [3,5,6,7,8]. These barriers safeguard tissues through the presence of tight junction (TJ) barriers between neighboring cells and the restrictive transcellular and/or paracellular diffusion of various molecules. In turn, TJ barriers are under tight physiological control by signaling pathways that influence the remodeling of the barrier (i.e., stabilization, opening, closing) [9,10]. Among several other protein kinases, mTOR (mTORC1/TROC2) and focal adhesion kinase (FAK) (p-FAK-Tyr397/p-FAK-Tyr407) have been shown to influence the integrity of the barriers restructuring the cell–cell junctions [11,12,13,14]. Of note, the presence of drug transporters (e.g., P-glycoprotein (ABCB1), breast cancer resistance protein BCRP (ABCG2), multidrug resistance protein 2 (MRP2) (ABCC2), multidrug and toxin extrusion protein 1 (MATE1) (SLC47A1)) at the BTB also influences the delivery of nanoparticles to the tumor (reviewed in [15]).

To overcome the BTB and increase the tumor retention of nanoparticles, one of the plausible approaches could be based on the decoration of the NP surface with cell-penetrating peptides (CPPs). In the current review, we assess the application of CPPs in nanotechnology. In distinction to other relevant reviews on this topic [16,17,18,19,20,21], we specifically focus on current chemical and physical approaches to increase the permeability of the BTB to facilitate the transport of anticancer functionalized nanoparticles.

## 2. Functionalization of Nanoparticles by Cell-Penetrating Peptides for Crossing Histohematic Barriers

Cell-penetrating peptides (CPPs) constitute a group of different short peptides with a maximum of 40 amino acids that can pass through tissue and cell membranes with energy-dependent or -independent mechanisms. The main feature of CPPs is their ability to penetrate the cell membrane at low micromolar concentrations in vitro and in vivo without causing significant membrane damage [22]. These peptides are capable of delivering electrostatically or covalently attached biologically active cargoes to the cell with high efficiency and low toxicity.

CPPs were proposed in the 1980–1990s by the discovery of the transactivator of the transcription (TAT) protein of human immunodeficiency virus type 1 (HIV-1) [23] and 16 amino acid peptide penetratin, derived from the homeobox Antennapedia transcription factor of *Drosophila melanogaster* [24,25]. Today, there are more than a thousand CPPs with different origins and structures; however, TAT and penetratin are the most studied. The great interest in CPP investigation has motivated researchers to develop a CPP database that provides comprehensive information on experimentally validated peptides (Database URL: http://crdd.osdd.net/raghava/cppsite/) [26,27] (accessed on 1 February 2023).

### 2.1. Classification of CPPs

The presence of a wide range of CPPs has contributed to their classification based on the origin of peptides, structure, main function, and cell penetration mechanism. The most relevant is the systematization of peptides into three groups according to their physicochemical properties as follows: hydrophilic (cationic), amphipathic, and hydrophobic peptides.

Hydrophilic peptides (e.g., TAT, penetratin, R8, etc.) are a class of CPPs with a high positive net charge at physiological pH due to the significant amount of lysine and arginine residues in their structure. These cationic amino acids mediate the electrostatic interactions of CPPs with negatively charged groups on the cell membrane. Moreover, the number and order of positive amino acids in a peptide sequence are critical factors for abundant cellular uptake. Arginine residues have been shown to be more favorable for cellular uptake than lysine because they contain a guanidine head group that more easily forms hydrogen bonds with phosphates and sulfates of the cell surface, leading to better CPP internalization [28].

Amphipathic peptides (e.g., SynB, RGD, etc.) are defined as peptides containing both polar (hydrophilic) and nonpolar (hydrophobic) regions having a different distribution in their structure. Besides lysine and arginine, these CPPs are also rich in hydrophobic residues, such as leucine, isoleucine, proline, and valine. Amphipathic peptides are divided into primary, secondary, and proline-rich CPPs [29]. Primary peptides are regions of natural proteins or chimeric peptides obtained by covalently binding a hydrophobic domain that promotes efficient targeting to the cell membrane with a nuclear localization signal (NLS). Secondary amphipathic peptides generally have a peculiar α-helical or β-sheet structure consisting of hydrophobic and hydrophilic amino acid residues grouped on opposite sides. This organization of the peptide allows it to orient in a specific way relative to the membrane surface, resulting in increased affinity and enhanced cellular uptake [30]. Another interesting class of amphipathic peptides is proline-rich CPPs, which have a heterogeneous structure and amino acid sequence, but all contain a proline–pyrrolidine template [31].

CPPs containing only nonpolar residues (e.g., C105Y, PFV, Pep-7, etc.) are termed hydrophobic peptides. These peptides have a low net charge and a high cell membrane affinity, derived from natural proteins or chemically modified. Due to hydrophobic residues, peptides are able to spontaneously translocate across membranes in an energy-independent manner, thereby exhibiting behavior that differs from the other classes of CPPs [32].

### 2.2. Mechanisms of CPP Uptake

Although CPPs rapidly penetrate the cell, the details of this mechanism are still not clear. This is due to the particular properties of peptides, their used concentration, the type of attached cargo, and the target cell. Currently, two groups of peptide penetration mechanisms have been proposed: (i) energy-independent direct translocation and (ii) energy-dependent endocytosis (Figure 1). It should be noted that the mechanisms of peptide release from the cell (for example, when crossing physiological barriers) have not actually been investigated. Based on the analysis of literature data, we speculate that the release of CPPs from the cell can occur through an energy-independent mechanism of direct translocation due to membrane destabilization.

#### 2.2.1. Energy-Independent Direct Translocation

Energy-independent direct translocation is a diffusion of CPPs through the lipid bilayer, controlled by the membrane potential. It involves several mechanisms based on the primary interaction of the positively charged amino acid residues of the peptide with the negatively charged membrane components. This is followed by the entry of CPPs into the cell cytoplasm, avoiding organelle confinement. Thus, in drug delivery, molecules that prefer direct translocation can avoid endosomal uptake [33]. The most studied methods of direct penetration include inverted micelle formation, the pore formation model, and the carpet-like model. 

Inverted micelle formation. The mechanism of inverted micelle formation was first proposed in the study of penetratin internalization [34]. At the site of interaction between the peptide and the lipid bilayer, the membrane destabilizes with the formation of a negative curvature. Invagination leads to the rearrangement of nearby lipid molecules and the formation of an inverted micelle encapsulating the peptide. The contents of the micelle are then released into the cell cytoplasm.

Pore formation model. The accumulation of CPPs near the outer leaflet of the plasma membrane can lead to thinning of the bilayer due to the interaction of positive peptide residues with lipid phosphate groups. This deformation contributes to the formation of transient toroidal or barrel-stave pores through which peptides diffuse into the cell [35]. This model occurs when the peptide concentration exceeds the concentration threshold specific for each particular peptide [36]. 

Carpet-like model. The first step of the carpet-like model is the parallel alignment of positively charged peptide segments relative to the lipid bilayer. Then, there is a spatial rearrangement of the peptide and a violation of lipid packaging, after which the peptide penetrates into the cell. 

All these models of direct penetration cannot accurately explain how CPPs deliver cargoes 100-fold their own size into the cell.

#### 2.2.2. Energy-Dependent Endocytosis

In addition to the electrostatic interaction of CPPs with lipid molecules and the formation of hydrogen bonds, which are responsible for the direct penetration of CPPs or CPP-cargo through the membrane, endocytosis remains the main delivery route. It includes two phases: (i) endocytotic entry and (ii) endosomal escape. Depending on the origin and physicochemical properties of the peptide and the cargo type, endocytotic entry can occur in several pathways: macropinocytosis, clathrin-mediated endocytosis, and caveolae/lipid raft-mediated endocytosis [37].

Macropinocytosis is accomplished by the inward folding of a small area of the outer surface of the cell membrane, resulting in the formation of a vesicle called a macropinosome. Subsequent membrane invagination occurs in the presence of the dynamin protein. Arginine-rich peptides are known to be internalized into the cell via micropinocytosis [38]. This has been shown using the inhibitor 5-(N-ethyl-N-isopropyl) amiloride (EIPA) and the F-actin polymerization inhibitor cytochalasin D. Moreover, arginine CPPs are able to cause rearrangement of the actin cytoskeleton, similar to that seen during micropinocytosis.

Clathrin- and caveolae-mediated endocytosis. These mechanisms of endocytosis occur with the participation of clathrin and caveolin pits, respectively. The proteins clathrin and caveolin localize to the inner side of the membrane and are required for membrane invagination and vesicle formation. Clathrin-mediated endocytosis is involved in the uptake of penetratin, the Tat peptide, and other CPPs after inhibition by the hyperosmolar medium [39]. The application of inhibitors of the clathrin-dependent pathway affects the CPP penetration level [40]. Caveolae-mediated uptake was also demonstrated for some CPPs (proline-rich CPPs, transportan) using inhibitors chlorpromazine (CPZ), methylated-β-cyclodextrin (MβCD), EIPA, and LY294002 [41]. 

The second phase of energy-dependent endocytosis is endosomal escape, which avoids the degradation of the cargo, allowing it to reach the target cell compartment and demonstrate its biological activity. Currently, there are several hypotheses to explain the endosomal escape pathways; however, the exact mechanism remains elusive. One hypothesis is based on the interaction of positively charged peptides with negatively charged phospholipids of the endosomal membrane, which leads to a change in membrane stiffness, pore formation, and the leakage of vesicle contents [42]. Another possible mechanism points to the effect of a pH gradient across the endosomal membrane, since acidic pH enhances the interaction of CPPs with the membrane and their transduction. In addition to the proposed hypotheses, specific agents have been developed to increase the effectiveness of endosomal escape. To increase the permeability of the endosomal membrane, pH-sensitive lipid molecules and peptides are used, which are additionally attached to the CPP-cargo complex [33]. Buffering agents (e.g., histidine, polyethylenimine (PEI)) are also used to increase osmotic pressure within endosomes, leading to their swelling, rupture, and release of contents [43,44].

### 2.3. Penetration of CPPs through Physiological Barriers

Currently, more than a thousand CPPs of various structures and origins have been investigated. All of them are applied for the targeted delivery of cargoes to the corresponding cells, tissues, and organs. Due to the presence of physiological barriers in the body, such as the blood–brain barrier (BBB), the blood–ocular barrier (BOB), the blood–thymus barrier, etc., the delivery of large molecules is severely limited, so the use of CPPs opens up new possibilities for theranostics of human diseases. Based on the analysis of experimental literature on CPPs, we have selected several promising candidates for the delivery of cargoes via the BTB. Herein, we have described the main characteristics and methods of the application of CPPs, such as TAT, SynB1, penetratin, R8, RGD, and angiopep-2 (Table 1).

#### 2.3.1. TAT Peptide

TAT is a transcriptional activator protein encoded by HIV-1 [23]. It belongs to a class of protein-derived peptides responsible for its translocation ability. TAT(47–57) peptide is derived from the core domain of the 86-mer TAT protein and has a basic YGRKKRRQRRR motif. TAT is a cationic CPP with a total charge of +8. The mechanism of cellular uptake of the TAT peptide is influenced by the physicochemical characteristics of the cargo and its experimental conditions (concentration of CPP, temperature, culture medium, cell type). It is assumed that the TAT peptide at low concentrations (<10 μM) passes through the cell membrane by direct penetration, but when the peptide is conjugated with a cargo, the mechanism of energy-dependent clathrin-mediated endocytosis is activated [45].

TAT is one of the most highly investigated CPPs for drug delivery across physiological barriers. Due to its striking penetrative properties, TAT can be used for the diagnosis and treatment of neurodegenerative diseases (i.e., Alzheimer’s and Parkinson’s), cerebral ischemia, malignant tumors, eye diseases, etc. Moreover, TAT can be administered via various clinical routes, including oral administration, injection (intravenously, subcutaneously, and intramuscularly), the transdermal route [46], the transmucosal route, and as a nasal spray [47]. The biodistribution of the TAT peptide labeled with [99mTc] was determined by gamma scintigraphy following tail vein bolus injection into Balb/c mice [48]. Biodistribution analysis revealed that the conjugate reached peak organ levels within the first few minutes after injection and displayed modestly rapid blood clearance. The maximum accumulation of TAT in the brain was observed 5 min after the injection and amounted to 0.39%ID/g. Over the subsequent 2 h, the conjugate was cleared by both renal and hepatobiliary excretion.

To increase the target properties of TAT, it is combined with various specific proteins and antibodies. For example, the protein Bcl-xl, which suppresses apoptosis in brain neurons, was conjugated with TAT [49]. The TAT-Bcl-xl fusion protein was distributed into various parts of the brain and caused protection against ischemia after intraperitoneal (IP) administration in animals. The conjugation of the TAT with pituitary adenylate cyclase-activating polypeptide (PACAP) facilitated the passing of PACAP-TAT through the BBB, the blood–air barrier (BAB), and the blood–testis barrier with efficiency about 2.5-fold higher than that of PACAP [50]. TAT has also been used to deliver vasoactive intestinal peptide (VIP) and PACAP across the BOB of rats via topical eye drop administration for its retinoprotective effect [51]. Conjugates VIP-TAT and PACAP-TAT reached the retina with an efficiency about threefold that of the VIP and PACAP.

#### 2.3.2. SynB1 Peptide

SynB1 is an 18-mer linear peptide derived from protegrin-1, which is an antimicrobial peptide found in porcine leukocytes. SynB1 translocates through biological membranes with high efficiency and is used for delivering molecules across the BBB for the treatment of brain diseases. For example, a study of brain uptake of dalargin (Dal), a hexapeptide analog of Leu-enkephalin that is not able to cross the BBB, in combination with SynB1 showed that the distribution volume for Dal-SynB1 was 18-fold higher compared with a free drug using in situ brain perfusion in mice [52]. Intravenous (IV) administration of doxorubicin (DOX) conjugated with SynB1 (DOX-SynB1) led to a significant increase in brain DOX concentrations during the first 30 min compared with free DOX [53]. Moreover, certain tissues, such as the heart, lungs, and, to a lower extent, kidneys and liver, had a lower uptake of DOX-SynB1 than free DOX. In addition, SynB1-conjugated gelatin–siloxane NPs (SynB1-PEG-GS) selectively accumulated in the brain of mice after IV administration compared to the nonconjugated PEG-GS NPs [54]. This confirms that SynB1 is able to cross the BBB and has significant potential for brain targeting and drug delivery.

#### 2.3.3. Penetratin

Penetratin is a peptide derived from the Antennapedia homeodomain of *Drosophila melanogaster* [24]. This homeodomain is a transcription factor (60 amino acid sequence) that binds to the DNA and is structured in three α-helices. The third helix of the homeodomain is involved in the translocation process and recognized as a penetratin peptide (residues 43–58) with the sequence RQIKIWFQNRRMKWKK [25]. The cellular uptake mechanism of penetratin is energy-dependent endocytosis [55].

The half-lives of penetratin, TAT, and SynB1 conjugated to the chelator DOTA and radiolabeled 111In (111In-DOTA-CPP) in human serum (t1/2) are 1.2, 8.8, and 5.2 h, respectively [56]. The difference in degradation rates may be related to the number of arginine–arginine bonds in peptides. The efficiency of uptake by different tumor cell lines (i.e., human anaplastic thyroid carcinoma SW1736, human prostate cancer PC-3, human colorectal carcinoma HCT-116, Morris hepatoma cells MH3924A, human breast cancer MCF 7, human squamous cell carcinoma HNO 97) decreased in the series penetratin, TAT, and SynB1. SW 1736 showed the highest accumulation rate (186.6% applied dose/106 cells after 30 min for penetratin), and HCT 116 showed the lowest accumulation rate (9.3% applied dose/106 cells after 30 min for SynB1). The in vivo biodistribution of CPPs in PC-3 tumor-bearing mice revealed accumulation in well-perfused organs, including the liver, spleen, lung, and kidneys. CPPs displayed relatively fast blood clearance. Penetratin, TAT, and SunB1 were found to cross the BBB within 10 min at the levels of 0.9%ID/g, 0.4%ID/g, and 0.3%ID/g in the brain, respectively.

Penetratin has been successfully applied for delivering proteins, chemical molecules, nucleic acids, and siRNA to cells with different origins. The use of penetratin as a vector for the delivery of DOX to the brain of rats showed that the accumulation of DOX–penetratin in the brain parenchyma of rats was about 20-fold higher than the free drug [53]. The conjugation of penetratin with transferrin (Tf) and DOX-loaded liposomes promoted the efficient cellular transport of the encapsulated drug (approximately 90–98%) and maximum translocation of the drug across the brain–endothelial barrier (approximately 15% across in vitro and 4% across in vivo BBB in rats) [57].

Penetratin has also been shown to be a penetration enhancer for drug delivery to the fundus oculi via eye drop instillation [58]. The peptide demonstrated high cellular uptake and low toxicity to ocular cells and tissues, even at high concentrations, compared to TAT and R8. After being instilled into the conjunctival sac of rat eyes, fluorophore-labeled penetratin displayed a rapid and wide distribution in both the anterior and posterior segments of the eye, and could be observed in the corneal epithelium and retina lasting for at least 6 h.

#### 2.3.4. R8 Peptide

Oligoarginines are chemically synthesized CPPs containing different numbers of arginine residues, typically 4–12 amino acids. One of the promising peptides of this class is octaarginine, which consists of eight amino acid residues of L-arginine (R8) or D-arginine (r8). Octaarginines R8 and r8 are used to deliver nanoparticles of various compositions, nucleic acids, and chemotherapeutic drugs into cells and tissues via direct penetration and endocytosis mechanisms [59,60].

Analysis of the biodistribution of fluorescent-labeled R8 and r8 peptides in HeLa tumor-xenografted nude mice after IV administration revealed that r8 accumulation in the tumor xenografts was almost threefold than that of R8 and sixfold than that of the TAT and penetratin peptides [61]. All peptides showed relatively high accumulation in the kidney, liver, and lung. r8 was also conjugated with DOX to evaluate its in vivo anticancer activity. The administration of r8-DOX and free DOX resulted in a ~50% drop in tumor proliferation, although free DOX led to a ~20% loss in bodyweight in contrast to r8-DOX. 

#### 2.3.5. RGD Peptide

The RGD sequence (Arg-Gly-Asp) has been recognized as the minimal integrin sequence present in many natural ligands binding the αvβ3 receptor as fibrinogen, fibronectin, laminin, osteopontin, etc. [62]. Currently, RGD is the basic motif for a variety of molecules designed for binding to αvβ3 integrin and other integrins. Due to the conservation of the RGD sequence, the affinity of natural and chemically synthesized ligands is affected by other amino acid residues flanking the RGD motif. Besides the direct interactions between these residues and the integrin receptor, flanking groups influence the folding of the peptide and thereby the conformational features of the RGD motif. 

Cyclization is commonly used to improve the binding properties of RGD peptides. The main drawback of linear RGD peptides is their low binding affinity (IC50 > 100 nmol/L), lack of specificity (αvβ3 vs. αIIBβ3), and rapid degradation by proteases in serum [62]. Cyclic RGD peptides display a higher activity compared to linear RGD peptides. Since cyclization confers rigidity to the structure, it significantly increases the selectivity of the RGD sequence for a particular integrin subtype and resistance to proteolysis [63]. 

Analysis of the biodistribution of the labeled RGD peptide showed that [99mTc]-HYNIC-RGD was rapidly cleared from blood circulation and excreted through the kidneys [64]. The tumor uptake of the peptide was high and homogeneous for αvβ3-positive cell lines (1.94 ± 0.26%ID/g), compared with αvβ3-negative cell lines (0.06 ± 0.01%ID/g). Analysis of the biodistribution of 188Re-RGD peptides after intravenous administration to mice bearing the S180 tumor showed that the tumor uptake of 188Re-HGRGDGRGDF(D) (7.03 ± 1.18%ID/g and 5.54 ± 0.70%ID/g at 1 and 4 h post-injection, respectively) was much higher than that of 188Re-HGRGDF(D) (3.23 ± 0.72% ID/g and 2.91 ± 0.37%ID/g at 1 and 4 h post-injection, respectively) [65]. Better results of 188Re- HGRGDGRGDF(D) compared to 188Re-HGRGDF(D) for tumor targeting might be explained by the twofold presence of RGD sequences in this peptide, which provides a higher density of activated binding sites. The biodistribution of the complex with cyclic RGD at 1 h after IV injection of [67Ga]Ga-DOTA-c[RGDf(4-I)K] in U-87MG tumor-bearing mice indicated high tumor uptake (about 4.5%ID/g) and low uptake in other nontarget tissues, except the kidneys and liver and the intestine for excretion [66].

#### 2.3.6. Angiopep-2 Peptide

Angiopep-2 (TFFYGGSRGKRNNFKTEEY) is a 19-mer peptide derived from the Kunitz domain that binds to low-density lipoprotein-receptor-related protein-1 (LRP1) and crosses the BBB [67]. LRP1 is highly expressed in brain–endothelial cells and in brain tumor cells, neurons, and astrocytes [67]. Angiopep-2 shows great transcytosis ability and parenchyma accumulation. Angiopep-2 has been conjugated to a wide range of therapeutic agents for the treatment of brain diseases, including cancer, brain injury, stroke, epilepsy, and Alzheimer’s and Parkinson’s diseases. For example, the complex of angiopep-2 with anti-HER2 monoclonal antibodies (ANG4043) led to the improved accumulation of the drug and, as a result, a twofold increase in the survival rate of mice with intracranial tumors [68]. Angiopep-2 was also conjugated with DOX (ANG1997) and etoposide (ANG1009) [69].

ANG1005 is a novel engineered peptide (EPiC) that consists of three paclitaxel residues linked to angiopep-2. The brain uptake of ANG1005 measured by an in situ rat brain perfusion assay showed an 80-fold greater uptake than free paclitaxel [70]. ANG1005 showed good tolerance in Phase I clinical studies and reached Phase II for the treatment of recurrent high-grade glioma (NCT01967810). ANG1005 also reached Phase II clinical trials for breast cancer with recurrent brain metastases (NCT02048059).

**Table 1 nanomaterials-13-01140-t001:** Representative CPPs for crossing histohematic barriers.

CPP	Sequence	Origin	Class	Proposed Mechanism of Cellular Uptake	Biodistribution	**Physiological Barriers**	**Refs**
TAT_(47–57)_	YGRKKRRQRRR	HIV-1 TAT protein	Cationic	Direct penetration	[^99m^Tc]-TAT: 0.39 ± 0.05%ID/g in brain 5 min after tail vein bolus injection to Balb/c mice (gamma scintigraphy, Mean ± SEM) ^111^In-TAT: 0.9 ± 1.1%ID/g in tumor 10 min after intravenous injection to PC-3 tumor-bearing mice (measurement of radioactivity on a γ counter, Mean ± SD)	BBB, BTBB, BOB, BAB, BTB	[48,53]
SynB1	RGGRLSYSRRRFSTSTGR	Protegrin-1	Amphipathic	Direct penetration, endocytosis	^111^In-SynB1: 0.3 ± 0%ID/g in tumor 10 min after intravenous injection to PC-3 tumor-bearing mice (measurement of radioactivity on a γ counter, Mean ± SD)	BBB, BTBB, BOB	[53]
Penetratin	RQIKIWFQNRRMKWKK	Antennapedia *Drosophila melanogaster*	Cationic	Direct penetration, endocytosis	^111^In-penetratin: 0.4 ± 0.1%ID/g in tumor 10 min after intravenous injection to PC-3 tumor-bearing mice (measurement of radioactivity on a γ counter, Mean ± SD)	BBB, BTBB, BOB	[53]
R8	RRRRRRRR	Chemically synthesized	Cationic	Endocytosis	Alexa660-R8: 3.7 × 10^9^ photon/s/g 24 h after intravenous injection to tumor-xenografted mice (measurement of fluorescent intensity by IVIS Spectrum System)	BBB, BOB	[61]
RGD	RGD	Site of integrin αvβ3	Amphipathic	Endocytosis	^188^Re-HGRGDGRGDF(D): 7.03 ± 1.18%ID/g in tumor 1 h after intravenous injection to S180 tumor-bearing mice ^188^Re-HGRGDF(D): 3.23 ± 0.72%ID/g in tumor 1 h after intravenous injection to S180 tumor-bearing mice [^67^Ga]Ga-DOTA-c[RGDf(4-I)K]: about 4.5%ID/g in tumor 1 h after intravenous injection to U-87MG tumor-bearing mice	BBB, BTBB, BOB	[65,66]
Angiopep-2	TFFYGGSRGKRNNFKTEEY	Kunitz-derived peptide	Amphipathic	Endocytosis	Cyto750-ANG4043: about 42 × 10^3^ p/s/mm^2^ in tumor 24 h after intravenous injection to BT-474 tumor-bearing mice (measurement of net fluorescent intensity by NiR imaging)	BBB, BTBB,	[68]

Abbreviations: BBB—blood–brain barrier; BAB—blood–air barrier; BOB—blood–ocular barrier; BTBB—blood–tumor brain barrier; BTB—blood–tumor barrier.

In conclusion, application examples of the peptides described above allow us to conclude that CPPs are a powerful tool for enhancing the penetration of molecules through the cell membrane. However, despite the significant advantages, there are also some limitations that prevent the formation of a multipurpose peptide for the delivery of diagnostic or therapeutic molecules to a specific area of the body. Firstly, many peptides have low cell, tissue, and organ specificity that require their local injection into the target tissue. Secondly, the attachment of a cargo to the CPP affects the cellular uptake of the conjugate, its physicochemical characteristics, and application efficiency. Thirdly, all CPPs and their complexes with cargo have different cytotoxicities for different cell types. The stability of the complex in physiological fluids plays an important role during IV administration of the conjugate, which in some cases may require modification of the peptide. For the application of CPPs in the theranostics of tumors and overcoming the BTB, the determining factor is the etiology of the tumor and its localization, since the degree of CPP accumulation directly depends on the location of the target site. Based on the analyzed data on the biodistribution of peptides, we assume that CPP RGD in various modifications and angiopep-2 have the highest ability to reach their target. However, to effectively assess the use of peptides to overcome the histohematic barrier, a detailed selection of experimental conditions is required taking into account all limiting factors.

### 2.4. Application of NPs Decorated with CPPs for Tumor Theranostics

The official definition of NPs given by the European Commission is that of “an organic or inorganic object with a dimension in the range from 1 to 100 nm”. NPs are considered colloidal carriers with a size between 1 and 1000 nm in terms of biology. There are many types of NPs of varying shape, size, surface charge, composition, and functionality. NPs can be classified as organic (e.g., lipid, polymeric NPs, etc.) and inorganic NPs (e.g., metal, ceramic, etc.). Each of them has advantages and disadvantages for tumor theranostics. Particle characteristics affect the interaction of NPs with cells, their accumulation in tissues, and their passage through various physiological barriers. The efficiency of drug delivery is determined by the combination of NP properties, which is of high importance in the process of particle synthesis, modification, and functionalization.

#### 2.4.1. Lipid-Based Nanoparticles 

Lipid-based nanoparticles (LNPs) have a number of unique properties and represent one of the most promising nanocarriers for drug delivery and tumor theranostics. LNPs can be classified as liposomes (LSs), solid lipid nanoparticles (SLNPs), and nanostructured lipid carriers (NLCs) and have many advantages, including ease of composition, self-assembly, biocompatibility, high bioavailability, ability to carry a large payload and range physical, and chemical properties that can be controlled to modulate their biological characteristics [71]. LSs are one of the most studied delivery systems due to the biocompatibility and biodegradability that they present. The main components of the LSs are phospholipids, which are organized in bilayer vesicles due to their amphipathic properties. Thus, they are able to encapsulate both hydrophobic and hydrophilic drugs [72]. Cholesterol-modified LSs increase the permeability of hydrophobic drugs through a bilayer membrane that improves their stability in the blood [72]. However, major disadvantages of LSs include the lack of available preparation methods, low degree of drug loading capacity and stability, and rapid degradation in the human body until the therapeutic effect can be achieved. SLNPs are colloidal drug delivery systems composed of physiological lipids, which remain solid both at room and body temperatures. The solid lipid forms a matrix material for drug encapsulation and is a stabilized mixture of surfactants or polymers. They have low toxicity, physical stability over a long period and control of release of drugs, site-specific targeting, easy large-scale production, and simple sterilization [73]. NLCs are nanocarriers developed from SLNPs, which represent a combination of solid and liquid lipids. The advantage of these systems is the higher ability to encapsulate a wide range of drugs soluble in liquid and solid lipid phases [74]. 

There are two main ways to deliver nanocarriers. One of them is passive targeting, which occurs when liposomes penetrate into tumor cells only due to molecular movement through the cell membrane (Figure 2). The passive targeting of liposomes to tumor tissues happens mainly due to the different pore sizes between endothelial cells of the tumor microvasculature compared to the ‘tighter’ structures found in normal capillaries. Therefore, ideal liposomes should be of a size that allows them to penetrate into tumor tissues, and not into normal tissues. 

Several studies have reported that the passage of NPs through the BBB is inversely proportional to the size [75].

Another route for NPs passing the barrier is active targeting, in which NPs are modified with antibodies or CPPs that recognize tumor cells (Table 2). NPs employ both types of transcytosis to cross the blood–tissue barriers: adsorption-mediated transcytosis (AMT) using a lectin-dependent mechanism and receptor-mediated transcytosis (RMT) using such surface receptors as the transferrin receptor (TfR) and the low-density lipoprotein-receptor-related proteins (LRP) by clathrin- or caveolo-dependent mechanisms [76]. In order to increase NP biocompatibility, several coatings of the particles were proposed. Thus, a polyethylene glycol (PEG) coating could minimize unwanted interactions with normal tissues and reduce NP clearance by the liver, spleen, and macrophages [77].

Nanoparticle surface modification with CPPs significantly improved tumor intracellular uptake, efficiency of barrier passing, and drug delivery, devoting to better curative effects and minimized side effects by reducing therapeutic doses. The attachment of CPPs to NP surfaces can be performed via electrostatic interactions or through covalent coupling strategies [21].

#### 2.4.2. Polymer-Based Nanoparticles (PNPs)

Over the past two decades, tremendous advances have been made in areas of the biomedical application of biodegradable polymeric materials, in particular as controlled/stable drug-carrying devices. Polymers are the largest group of biomaterials with unique properties such as toughness, flexibility, and mechanical strength [78]. The diversity of the composition of polymer nanoparticles is vast, including monopolymers, copolymers, lipid or metal–polymer hybrids, etc. [79,80]. Despite the fact that polymers can be referred to as excipients, they are able to change the biopharmaceutical and biokinetic properties of the transported active molecule, increasing its efficiency and stability, as well as reducing cytotoxicity in relation to healthy peripheral tissues [81]. PNPs can be synthesized from natural or synthetic materials, as well as monomers or preformed polymers, thus allowing a wide variety of possible structures and characteristics. The most common type of PNPs includes gelatin, chitosan, polylactic acid (PLA), polycaprolacton (PCL), polylactic-co-glycolic acid (PLGA), and others. PNPs have been also investigated as biocompatible and promising nanocarriers to deliver drugs across the different physiological barriers. 

It was shown that PNPs coated by polyethylene glycol (PEG) and decorated with CPPs can successfully overcome BBB. Particularly, angiopep-modified NPs were developed as a promising brain-targeting nanocarrier for lipophilic drugs and a theranostic platform for glioblastoma (Table 2) [82]. Among the important factors affecting the therapeutic efficacy of targeted polymeric nanoparticles are ligand density and linker length. It was shown that the combination of short PEG2k linkers and high cRGD surface modification of PLGA synergistically increased nanoparticle uptake by glioblastoma cells (Table 2). The optimal ligand value for increasing cellular internalization depends on the increase in receptors on the cell surface. However, as ligand density increases, the increase in binding affinity may actually decrease exocytosis efficiency [83]. Thus, the balance between ligand density and receptor density is an important key to the NP transport across the BBB. 

Despite all the advantages, natural polymers also have limitations in medical applications, including insolubility in water and many other organic solvents and low viscosity [84]. Another disadvantage of polymeric NPs is an increased risk of aggregation and toxicity. Only a small number of polymer nanodrugs (e.g., Eligard (Tolmar) and Abraxane (Celgene)) are currently approved by the FDA for cancer theranostics.

#### 2.4.3. Inorganic Nanoparticles 

In recent years, inorganic nanoparticles (INPs) have attracted much attention in the field of anticancer therapy due to the advantages of easy preparation, hypotoxicity, biosafety, and easy modification. These NPs have magnetic, radioactive, or plasmonic properties, which make them also suitable for diagnostics, visualization, and photothermal therapy. The most used materials for their synthesis are gold, iron, and silica. INPs have a precise formula and can be prepared with given sizes, structures, and geometries. Based on gold NPs (AuNPs), various forms are synthesized, such as nanospheres, nanorods, nanostars, nanoshells, and nanocells [85,86]. INPs include gold, iron oxide, calcium phosphate, and mesoporous silica and can be easily functionalized, providing them with additional properties such as drug delivery. Iron oxide NPs are the majority of FDA-approved inorganic nanomedicines [87,88]. Compared to gold and silver, superparamagnetic iron oxide nanoparticles (SPIONs) are much more economical and easy to synthesize. However, SPION colloidal instability is a factor that limits their therapeutic application. Given this limitation, various biocompatible materials are used for modification, including polyethylene glycol (PEG), dextran, poly(ethyleneimine) (PEI), etc. [89]. 

Recently, considerable attention has been paid to nanocarriers decorated with CPPs for tumor theranostics. Receptor-mediated transcytosis is one of the mechanisms to overcome NPs through tight junctions between endothelial cells and the delivery of therapeutic agents to tumor tissues [90]. INPs decorated with various CPPs (e.g., RGD, TAT, angiopep, etc.) overcome physical barriers and are absorbed by tumor cells (Table 2). 

Many characteristics of the tumor microenvironment, including the vasculature, interstitial fluid pressure, and extracellular matrix density, contribute to the limited entry and entry of NPs. The combination of several CPPs or specific receptor-targeting antibodies, as well as various materials in one multicomponent particle, allows the creation of nanoplatforms that overcome the limitations of single-component structures (Table 2) [68,69]. Advanced delivery methods, such as the application of hybrid NPs or surface modification of NPs (such as dense PEG coverage and CPP combination), have been explored [91,92]. These methods can help to improve NP penetration through the BBB and promote a more even distribution throughout the brain. Table 2 summarizes the in vitro/in vivo, in situ, and clinical studies carried out in the most frequent cancers employing CPPs functionalized NPs.

## 3. Chemical Modification of the BTB Permeability

Drug delivery systems can be chemically modified in a variety of ways to help them penetrate histohematic barriers more effectively: (1) osmotic agent administration; (2) inhibition of efflux pumps (e.g., P-glycoprotein inhibitors etc.); (3) disruption of tight junctions; (4) disruption of extracellular matrix (ECM); (5) NO-donors (Figure 3). Some examples of studies using such technologies are summarized in Table 3.

The use of nanoparticles (NPs) in conjunction with osmotic chemical agents such as mannitol can locally increase the osmotic pressure on the cell membrane due to the attraction of water molecules, which makes the cell membrane incoherent and more permeable [93]. Osmotic pressure has also been shown to impact the cytoskeleton of cells and the strength of intercellular connections. For instance, co-administration of the most common osmotically active substance, mannitol, results in the improved BBB dispersion of temozolomide (TMZ) [94]. Drug delivery via the blood–tumor barrier can be effectively accomplished by NPs based on or conjugated with mannitol, polyurea, and borneol [95,96,97]. However, intracarotid injection of osmotic agents may in some cases cause behavioral and electrographic seizures; supposedly, this side effect may be avoided if such an osmotic agent is combined with NPs [98].

Multidrug resistance (MDR) transporters are (ATP)-binding cassette (ABC) efflux transporters that push different chemical compounds out of the cell with variable specificity. Among them, the most studied are P-glycoprotein and breast cancer resistance protein (BCRP). Tumor cells tend to overexpress P-glycoprotein, so it becomes obstructive for drug delivery [99]. Nanomaterials such as d-Alpha-tocopheryl polyethylene glycol-1000 succinate (TPGS), poly(ethylene glycol)-poly(D, L-lactide) (PEG-PLA/PLGA), and poloxamer polymers (pluronics) can nonspecifically inhibit P-glycoprotein activity itself [100]. Specific P-glycoprotein inhibitors such as elacridar, tariquidar, zosuquidar, and DP7 can be combined with NPs to increase the accumulation in tumor cells [101,102,103,104,105]. Elacridar conjugated to NPs, in particular, shows its distinctive property in penetration of the skin barrier [101].

Endothelial layer permeability is largely determined by the density of intercellular junctions (i.e., tight and adhesive junctions, TJs, and AJs). Intracarotid administration of mannitol leads to the widening of intercellular junctions to ~40 nm via Src-kinase-dependent TJ degradation [106,107]. Currently, superselective intra-arterial cerebral infusion of mannitol is a technique for enhancing the delivery of bevacizumab and cetuximab through the BBB due to expanding TJs [108]. Sodium caprate is also an applicable agent for nonspecific TJ disruption [109,110]. Co-administration of small interfering RNAs (siRNAs) targeting claudin-5, one of the TJ proteins, helped the paracellular clearance of amyloid-β from the brain due to the inhibition of claudin-5 level expression [111]. There is currently no proof that NP conjugates have been used with substances that particularly disrupt TJ; however, there are some substances that can modify TJ architecture by interacting with TJ’s extracellular loops. For example, claudin peptidomimetics and clostridium perfringens enterotoxin (CPE) are potential candidates for conjugation with NPs [112,113].

In tumors such as pancreatic cancer, a critical role in drug delivery having the ability of NPs to pass through the dense extracellular matrix (ECM), which consists of such branching molecules as collagen, fibronectin, proteoglycan, and hyaluronic acid [114]. Thus, the conjugation of NPs with ECM enzymes such as hyaluronidase, collagenase, and matrix metalloproteinases (MMPs) is considered a bioavailability enhancer of main drugs in the therapy of pancreatic and breast cancers [115,116,117]. The straightforward composition of NPs, containing molecules such as hyaluronidase, results in the fast inactivation and biodegradation of NPs due to the serum proteins in blood flow. Involved in NPs, for example, additional PEG coating and sequestering hyaluronidase may drastically improve NP blood circulation time [115]. Using collagenase-conjugated NPs is not effective as hyaluronidase-conjugated NPs. Presumably, poor tissue clearance of collagenase products and bad time administration of collagenase-containing drugs may trigger metastasis [118,119]. 

Conjugating NPs with NO-donors is a novel strategy in antitumor therapy because of its multifunctional effects, including the induction of apoptosis and inhibition of metastasis, reversing the MDR effect via the inhibition of P-glycoprotein expression and improving drug delivery via the enhanced permeability and retention (EPR) effect by stimulating tumor angiogenesis. Moreover, NO-donors tend to be activated by endogenous (e.g., pH, GSH, H_2_O_2_, etc.) and exogenous (e.g., light, UV) stimuli [120]. The co-encapsulation of various forms of NO-donors with common drugs such as doxorubicin, paclitaxel, and irinotecan was found to be an effective antitumor strategy as NO-donors are antitumor agents by themselves and enhance the permeation of co-delivered drugs [121,122,123,124,125]. The most interesting feature of NO-donors is their ability to form gas in response to environmental stimuli, and their application is found in the engineering of so-called drug delivery systems (DDS), dynamically changing nanocarrier structures. One of the main challenges in NO-based anticancer therapies is to choose an accurate NO dosage. This problem is associated with an existing gap in the NO level, at which tumor progression can be either accelerated or inhibited [126].

Drug delivery systems, on the one hand, can provide specific local tissue activation of drugs and, on the other hand, can boost drug bioavailability in target cells. The composition of DDSs can be activated to release drugs by environmental stimuli such as low pH, redox [122,127] or tissue-specific esterases, enzymes, etc. [125,128]. Globally, DDSs can be separated into two types of structure: (i) drug-encapsulated NPs [122,125] and (ii) self-assembled NPs [127,129]. Drug release might happen extracellularly, as in [122], where NO-donors stimulate the degradation of the PEG shell of NPs in response to acidic microenvironments. Alternatively, the activation of NP dissociation may occur inside the cell [127,130] when self-assembled NPs from superhydrophobic Pt(IV)-6 and amphiphilic lipid-PEG enter the cells via macropinocytosis and then reassemble due to the high glutathione level for the release of the antitumor Pt(IV) drug. As MMP-9 has been found to be overexpressed in tumors, DDSs may be also accompanied within the MMP-9 cleavable linker to achieve tumor EMC-specific drug release [125]. Self-assembling prodrug NPs can be split into a separate category of DDSs. Such DDSs are composited mainly from prodrugs such as fatty-acid-linked irinotecan or amphiphilic glucosyl acetone-based ketal-linked etoposide glycoside prodrug [128,129], whose oligomeric form drastically changes their surface chemical characteristics and cell penetration ability.

A radically different approach in passing through the histohematic barrier is using cell-based systems as the drug delivery technique. Mesenchymal stem cells (MSCs) are the most potent cell line for this technique as they have endogenous tumor-homing activities. After local injection of MSCs, they can be found in the perifocal tumor zone, supposedly for tumor stroma remodeling. Using MSCs as a delivery system for NPs is controversial for their unpredictable effects [131]. However, there is still some successful evidence of the application of MSC-based NP delivery in vivo models [132,133,134].

In conclusion, there are different approaches in the chemical modification of NPs for the better penetration of histohematic barriers. However, for the appropriate choice of NP chemical composition, the individual tissue characteristics of targeted organs and other concomitant patient diseases should be considered. In the context of conjugation with NPs, applying TJ peptidomimetics and efflux pump inhibitors moieties looks less toxic among all the presented variants. The development of DDSs, especially in anticancer therapy, seems to be a promising step in the solution of the drug delivery issue, as it can theoretically significantly enhance the safety and effectiveness of NP application, although further preclinical studies are of high importance. 

## 4. Physical Methods of Histohematic Barriers Permeability Enhancement

There are several most frequently used histohematic barrier permeability enhancement techniques, which are based on the physical effect using NPs. These techniques include NP size reduction [135], NP surface charge modulation [136], NP shape modulation [137], magnetic-field-enhanced permeation [138], the effect of focused ultrasound [139], and the effects of electromagnetic radiation and electroporation [140]. The schematic illustration of the technique principles is shown in Figure 4. A comparative analysis of the techniques is given in Table 4.

The reduction of NP size makes it possible to implement both the paracellular pathway through tight junctions and passive transmembrane diffusion [135]. Various histohematic barriers are permeable for NPs with a size below approximately 10 nm for the BBB (e.g., yttrium oxide NPs [141], 2 nm [142], or 3 nm [143] AuNPs, polysiloxane NPs loaded with Gd chelate [144]), ~10–50 nm for the blood–brain tumor barrier (e.g., metallic NPs [145], NPs made of monomethoxy(polyethylene glycol)d,l-lactic-co-glycolic acid [146]), intestinal barrier (titanium dioxide NPs [147]) and blood–air barrier (e.g., fluorescent polystyrene nanospheres [148], AuNPs [16]), ~3 nm for glomerular barrier (e.g., AuNPs coated with glutathione [143], gold nanoclusters with a size less than 1 nm [149]) and blood–thymus barrier (nanoclusters of Mo132 and Mo_72_Fe_30_). In the case of BBB permeation, the size decrease can not only promote drug delivery to the brain but also reduce drug uptake by the liver, thus decreasing drug toxicity [150]. Despite the above-mentioned advantages of using ultrasmall NPs and atomic clusters, the prospect of their application in oncotheranostics raises certain doubts due to the need to use relatively large (about 100 nm) conjugates of NPs with drug molecules.

Depending on the surface charge, NPs can undertake adsorptive-mediated transcytosis, which is triggered by electrostatic interactions between the positively charged substrate surface and the negatively charged plasma membrane surface of the endothelial cells [136]. The use of the suitable surface charge allows larger NPs to cross barriers up to 200–300 nm for solid lipid NPs (Zeta potential value raging from +61 mV [151] to −18 mV [152,153]), emulsifying wax NPs (Zeta potential value constituted −14 mV for neutral, −60 mV for anionic, and +45 mV for cationic charged NPs [154]) and poly(lactic-co-glycolic acid) NPs (Zeta potential = + 58 mV [151]) crossing BBB; up to 100–150 nm for iron oxide NPs coated with cationic polyethylenimine and anionic carboxymethyldextran (Zeta potential absolute value is more than 24 mV) [155] and human serum albumin and lauric-acid-coated maghemite NPs (Zeta potential constituted −21 mV) crossing blood–placenta barrier. To date, there are no clear recommendations on the sign and magnitude of the NP Zeta potential used for transbarrier delivery. For this reason, the generally accepted approach can be considered to achieve the maximum possible absolute value of the Zeta potential for the drug used.

The modulation of the NP aspect ratio influences barrier permeability, and a higher aspect ratio usually corresponds to higher permeability [137]. As hypothesized in [156], rod-shaped NPs better transport across the BBB compared to their spherical counterparts because rods are trafficked through the cells using a route that is more efficient for transcytosis. Such differences could potentially involve fundamentally different pathways or simply enhanced efficiency of the same pathway for rod-shaped particles; therefore, the origin of the differences in routes needs further investigation. For the BBB, the appropriate maximum length of the rod-shaped particles is high enough (in comparison with the spherical NPs’ diameter). It reaches about 300 nm for polysterene [156,157] and mesoporous silica [158] particles with an aspect ratio of ~3:1 and about 40–60 nm for Au [159] and TiO_2_ [160] nanorods with an aspect ratio of about 2:1. The appropriate size of the star-shaped NPs is about 50 nm for Au [160,161] and about 100 nm for lipid NPs [162]. The use of NPs with a high aspect ratio, despite the increase in the permeability of histohematic barriers, in the opinion of the authors, cannot be considered practically useful. Since real complexes used in oncotheranostics usually have an isotropic shape close to spherical.

Magnetic field gradients can stimulate iron oxide NPs to cross the barrier without damaging it [138]. The source of the magnetic field can be a cylindrical rare-earth (NdFeB) magnet [163,164], including a magnet implanted subdermally [165] or placed under the culture plate [166,167], and a circular Halbach array composed of eight NdFeB magnets [168]. The strength of the applied magnetic field is reported in the range of 0.01–1.0 T [165,167,169]. Additionally, to the static magnetic field, the dynamic magnetic field produced by rotating NdFeB magnets (60 rpm) can be applied [168]. In addition, the typical NP size can be increased in comparison with the passive permeation and can reach about 100 nm for the BBB [166,170] and blood–spine barrier [165]. The enlarged NP diameter makes it possible to load the medicine, e.g., doxorubicin [163,165] and salinomycin [170], or target macromolecules such as the cell-penetrating peptide Tat [166]. The issues with toxicity occurring when magnetic NPs are degraded could be limited by the focused application of a magnetic field and thereby decreasing the amount of applied magnetic NPs [171]. The direction associated with the use of a constant magnetic field gradient, alone or in combination with other methods of barrier permeability modulation, seems to be one of the most promising due to the safety of a low-strength constant magnetic field relative to alternating magnetic fields and radiofrequency fields. However, it is difficult to solve the problem of creating a local magnetic field without highly invasive interventions, since the magnetic field strength of a point source decreases rapidly with distance.

The focused ultrasound creates reversible BBB permeability enhancement by concentrating acoustic energy on a focal spot and disrupting tight junctions [139]. The focused ultrasound intensity parameters can be specified in the units of the acoustic power (0.42–1.84 W [172,173]), power density (0.5–1.0 W/cm^2^ [174]), sound pressure (1000 kPa [175]), peak acoustic pressure (in the range of 0.09–0.80 MPa [173,176,177]), and focal acoustic pressure in the brain (0.8 MPa [178]). The ultrasonic transducer frequency also may be different and varies from 0.5 to 2.7 MHz [62,176,178,179]. A custom-built dual-frequency ultrasound transducer can be applied [177]. Depending on the ultrasound power and frequency, the maximum NP diameter reaches 100–200 nm for cell-membrane-cloaked liposomes [174], lipid–polymer hybrid NPs [172], and PEGylated Au NPs [176] and about 50–100 nm for polymeric micelles [177] and sulfur NPs [175]. Similar to the source of a magnetic field, the source of local ultrasonic impact will require prompt intervention in the area of interest, since for the impact of ultrasound, it is necessary to provide a continuous medium that conducts acoustic vibrations.

Reversible and irreversible electroporation may be achieved via the use of electric pulses delivered through needle electrodes inducing a nonthermal focal ablation to the target by a series of electric pulses [140]. Electrical pulse amplitude can reach 5–2000 V [180,181,182,183,184,185] at the pulse delivery frequency of 1 Hz or 4 Hz [186]. In the studies with the application of this technique, there are only few types of NPs are usually used for the barrier permeability visualization: Gd chelates (size is about 1 nm) ([183,184,186]), fluorescent dyes (size is about 1 nm) ([181,182]) for BBB and dye-stabilized Sorafenib NPs (size is about 100 nm) for the blood–tumor barrier [180]. Most of the studies reviewed by us, which describe the use of this method of increasing the permeability of barriers, are aimed at irreversible change of the barrier, usually with a brain tumor, in order to provide a drug effect on it. For this reason, electroporation can be considered a rather narrowly applicable approach, unsuitable for nonradical therapy, for example, for nonmalignant disorders.

Electromagnetic effects transiently increasing blood–brain and blood–tumor barrier permeability include magnetic heating of magnetic NPs with a low (magnetic field amplitude 7.6 kA/m at 150 kHz [187] and 33.4 kA/m at 300 kHz [188]) or high (source frequency 13.56 MHz at the power of 80 W [189], source frequency 915 MHz at the power of 5 mW [190] or 20 mW [191]) radiofrequency source, laser heating (green picosecond 532 nm laser with the doses of 2.5–25 mJ/cm^2^ [192], near-infrared (NIR) 980 nm laser with the power densities of 0.15–0.72 W/cm^2^ [193], near-infrared 808 nm laser with the doses of 10 and 30 J/cm^2^ [194] and femtosecond NIR laser with the power of 300–2000 mW [195]), ionizing radiation produced by the cone beam clinical source with the accelerating voltage of up to 6 MV at the doses of 2–10 Gy and laser/X-ray combined impact with the use of an NIR 808 nm laser at the power densities of 1–3 W/cm^2^ and a clinical X-ray source with the dose of 6 Gy [196]. If there is a favorable prognosis, the use of methods of this group seems undesirable due to possible delayed negative effects (for example, secondary tumors induced by radiation), as well as insufficient knowledge of the effects of low-intensity radiofrequency fields on viable tissues. 

Some other permeability enhancement techniques are described. These techniques include the combined effect of magnetic field and ultrasound [197], which can be applied for gene delivery by the use of mesoporous silica NPs with magnetite core loading microbubbles with a gas core. The microbubbles disrupt the barrier due to the cavitation effect and release NPs, which can directly pass through the barrier to the tumor tissue. Another promising technique is photoacoustic cavitation [198], which can be applied for tumor site-specific BBB opening for the delivery of therapeutic NPs in the photoacoustic therapy of glioblastoma. Photoacoustic cavitation provides reversible barrier opening due to thermal expansion and simultaneously mechanical damage of the tumor tissue [198]. Intracellular tension modulation [93] is a technique based on an increase in intracellular osmotic pressure, which enhances the barrier permeability and also causes upregulation of membrane fluidity, promoting nonselective drug influx. Protein nanoparticle-related osmotic pressure could be a novel therapeutic target for BBB lesion-related brain diseases and possibly the development of novel drugs that cross the BBB [93]. Summing up, we can say that today in real practice, the isolated use of the above physical methods for increasing the permeability of histohematic barriers is quite rare. The combined use of one or more physical methods and targeting molecules, for example, peptides, is more effective.

**Table 2 nanomaterials-13-01140-t002:** CPP-functionalized NPs for tumor theranostics.

NP Type	Peptide Type	Cancer/Barrier	Administration Route	Reference
Lipid-based nanoparticles				
LNPs	RGD	Gastric cancer	ivt + ivv	[199]
LNPs	TAT	Breast cancer	ivt	[200]
LNPs	TAT	GBM	ivt + ivv	[201]
LNP	Penetratin	GBM	ivt + ivv	[202]
LNPs	TAT	BBB	ivt + ivv	[203]
LNPs	R8	GBM	ivt + ivv	[204]
LNPs	RGD Angiopep-2	GBM	ivt + ivv	[205]
LNPs	Angiopep-2	GBM	ivt + ivv	[206]
LNPs (NLCs)	RGD	Gliomatosis cerebri	Clinical trial	[207]
Polymer-based nanoparticles				
PNPs	RGD	GBM	ivt + ivv	[208]
PM-Ch	TAT	BBB	ivt + ivv	[209]
PM	Angiopep-2	BBB	ivt + ivv	[210]
PLA	Angiopep-2	Brain tumor	ivv + evv	[82]
PLGA NPs	Angiopep-2	BBB	ivv	[211]
PLGA NPs	Penetratin	Cervical cancer	ivt	[212]
PLGA NPs	RGD	GBM	inv	[213]
PLGA NPs	TAT	Pancreatic cancer	ivt	[214]
Inorganic nanoparticles				
AuNPs	TAT	Lung carcinoma Breast cancer Colon cancer	ivt + ivv	[215]
AuNPs	RGD	GBM Breast cancer Melanoma	inv	[216]
AuNPs	RGD	GBM	in situ	[217]
AuNPs	RGD	Melanoma	inv	[218]
AuNPs	TAT	Breast cancer	ivt + ivv	[219]
AuNPs	R8 Angiopep-2	GBM	ivt + ivv	[220]
SPNPs	RGD	GBM	ivt + ivv	[221]
SPIONs	TAT	Nasopharyngeal carcinoma	N/A	[222]
MNPs	RGD	Breast cancer	ivt + ivv	[223]
Hybrid nanoparticles				
Lipid/PLGA nanocomplex	R8	Colon carcinoma	ivt + ivv	[224]
PEG-PLA NPs	Penetratin	BBB	ivt + ivv	[225]

Abbreviations: NPs—nanoparticles; LNPs—lipid NPs; AuNPs—gold NPs; PLGA NPs—poly (lactic-co-glycolic acid) NPs; PEG-PLA NPs—poly(ethylene glycol)-poly(lactic acid) NPs; MSNPs—mesoporous silica NPs; IONPs—Iron oxide NPs; SPIONs—superparamagnetic iron oxide NPs; SPNPs—synthetic protein NPs; PNPs—polymeric NPs; PM-Ch—polymeric micelles self-assembled from cholesterol; MNPs—melanin nanoparticles; Lipid/PLGA nanocomplex; ivt + ivv—in vitro/in vivo; GBM—glioblastoma multiforme; BBB—blood–brain barrier; R8—octaarginine peptide; N/A—not applicable.

**Table 3 nanomaterials-13-01140-t003:** Application of chemical effects to change the permeability of histohematic barriers.

Type of Chemical Modification	Chemical Modification	Chemical Composition of NPs	Coupling with NPs	Blood–Tissue Barrier	References
Osmotic action	Borneol	liposome encapsulation	+	BTBB	[226]
	Mannitol	Mannitol coupling camptothecin NPs	+	BTB	[95]
	Urea	- Glutamate-urea-based PSMA-targeted PLGA NPs - PEGylated Polyurea NPs	+	BTB (prostate cancer)	[96,97]
Efflux pump inhibitors (P-glycoprotein, ABC transporters inhibitors)	Elacridar	Tributyrin/oleic acid/tricaprylin Nanoemulsion	+	BAB	[101]
	Tariquidar	mSiO_2_-dPG nanocarriers; Nanoliposomes	+	BTB (gastric cancers)	[102,103]
	Pluronic L-61	Pluronic L-61/F127	+	BTB	[227]
	Zosuquidar	Nanoliposomes	+	BTB (liver cancer)	[104]
	DP7	Cholesterol-modified antimicrobial peptide DP7	+	BTB (hepatocellular carcinoma)	[10]
Tight junctions disruption	Claudin-5 peptidomimetics (C5C2)	-	-	BBB	[105]
	Sodium decanoate C_10_	-	-	Intestinal barrier; BBB	[109,110]
	*Clostridium* *perfringens* enterotoxin (CPE)	-	-	BAB	[113]
	siRNA against Claudin-5	-	-	BBB	[111]
ECM disruption	Hyaluronidase	PLGA-PEG	+	BTB (pancreatic cancer)	[115,228]
	Collagenase	PLGA-PEG-PLGA thermosensitive hydrogel	+	BTB (breast cancer)	[116]
	MMP-1	Glycerol monostearate/DSPE-PEG_5000_-Maleimid NPs	+	BTB (pancreatic cancer)	[117]
NO-donors	Nitroglycerin	Polyethylene glycol-conjugated zinc protoporphyrin IX	pretreatment	BTB	[229]
	S-Nitrosated human serum albumin dimer (SNO-HSA-Dimer)	N-(2-hydroxypropyl) methacrylamide polymer conjugated with zinc protoporphyrin PEGylated liposomal doxorubicin (Doxil)	pretreatment	BTB	[230]
	NO-donor conjugate	HPMA copolymer-bound cytotoxic drug (doxorubicin; Dox)	pretreatment	BTB	[231]
	Nitrate functionalized D-α-tocopherol polyethylene 1000 glycol succinate (TPGS)	TPGS-SS-PTX (paclitaxel) and TPGS-NO3 self-assemble hybrid micelles (TSP-TN)	+	BTB	[121]
	Diethylenetriamine diazeniumdiolate (DETA NONOate)	Irinotecan and DETA NONOate Co-incancapsulated in PLGA shell	+	BTB	[122]
	Hyaluronic acid with nitrate ester	BSA-protected gold nanoclusters	+	BTB	[123]
	S-nitrosothiols	CuS mesoporous silica core-shell nanocarrier	+	BTB (in vitro)	[124]
DDS	MMP-9- cleavable, collagen mimetic lipopeptide POPE-SS-PEG_5000_ polymer (GSH-sensitive)	MMP-9-cleavable, collagen mimetic lipopeptide micelle coated with POPE-SS-PEG_5000_ polymer (drug–gemcitabine)	+	BTB (pancreatic cancer)	[125]
	Glutation-responsive Pt prodrug	NPs from superhydrophobic Pt(IV)-6 and amphiphilic lipid-PEG	+	BTB	[127]
	Glucose oxidase activated DOX prodrug release	Doxorubicin prodrugs (pDOXs) with β-cyclodextrins (β-CDs) in spongy silica nanoparticle Pt^0^ nanoreactor (GPS-pDOX-CD)	+	BTB	[130]
	Cisplatin hydrate and Tolf cell release in response to acid environment.	Tolfplatin (Tolf and cisplatin hydrate) in hydrophobic lipid-Poly (lactic-co-glycolic acid) (PLGA) (Lipid-PLGA@Tolfplatin NPs)	+	BTB (breast cancer)	[232]
	Esterase-activeted irinotecan prodrug (SN38)	Micelle-forming macromolecular from SN38, conjugated with poly glutamic acid and polyethylene glycol hidrifilic segment	+	BTB	[233]
	Lysosomal release of etoposide prodrug	Self-assembling amphiphilic glucosyl acetone-based ketal-linked etoposide glycoside prodrug	+	BTB	[129]
Cell-based delivery system	Mesenchymal stem cells (MIAMI line)	Ferrociphenol lipid nanocapsules	+	BTBB	[132]
	Mesenchymal stem cells	Mesoporous silica nanoparticles	+	BTBB	[133]

Abbreviations: NPs—nanoparticles; BTBB—blood–tumor brain barrier; BTB—blood–tumor barrier; BAB—blood–air barrier; BBB—blood–brain barrier.

**Table 4 nanomaterials-13-01140-t004:** The histohematic barriers permeability enhancement with the use of the various physical techniques.

Barrier Permeability Enhancement Technique	Chemical Composition of NPs	NPs Dimensions (nm)	NPs Shape	Histohematic Barrier	Ref.
NP size reduction	Y_2_O_3_ core modified with poly(ethylene glycol methacrylate phosphate) and N-fluorescein acrylamide	7–8	Spherical	BBB	[141]
	Metallic	30–50		BTBB	[145]
	Au	8–12		BBB	[234]
	TiO_2_	22–45		Intestinal barrier	[147]
	Porous spherical nanocluster POMs Mo_132_ and Mo_72_Fe_30_	2.5–2.9		Blood–thymus barrier	[235]
	Fluorophore-conjugated Au cores	2		BBB	[142]
	Polysiloxane network with Gd chelates	5		BBB	[144]
	Poly(acrylic acid) stabilized Gd_2_O_3_ cores	8–13		BBB	[236]
	Fluorospheres^®^ (fluorescent polystyrene nanospheres)	50		BAB	[148]
	Au	4–7		BAB	[237]
	Fluorescence-labeled Au	10		BTBB	[238]
	Disulfiram in mPEG-PLGA matrix	70		BBB/BTBB	[146]
	TiO_2_	14–29		BBB	[239]
	Au core modified with glutathione	3		BBB/Glomerular barrier	[143]
	Au_18_, Au_15_, Au_10–11_, Au_25_ nanoclusters	<1		Glomerular barrier	[149]
NP surface charge modulation	Chitosan-coated insulin-loaded SLNs (ZP = +61 mV) and PLGA NPs (+58 mV),	139–151 and 165–186	Spherical	BBB	[151]
	Human serum albumin and lauric-acid-coated maghemite core (ZP = −21 mV)	97		BPB	[240]
	Neutral and positively charged fluorescent polystyrene nanospheres	22, 48, 100		BBB	[241]
	Iron oxide core coated with neutral starch (ZP = −11 mV), cationic polyethylenimine (+54 mV), and anionic carboxymethyldextran (−24 mV)	150		BPB	[155]
	Ag (ZP = −38 mV)	4–10		BBB	[242]
	Andrographolide-loaded SLNs (ZP = −30…−36 mV)	260–280		BBB	[153]
	Agomelatine-loaded SLNs (ZP = −18 mV)	167		BBB	[152]
	Neutral (ZP = −14 mV), anionic (ZP = −60 mV) and cationic (ZP = +45 mV) charged emulsifying wax NPs	20–200		BBB	[154]
NP shape modulation	Polystyrene rods (aspect ratio 2:1 and 5:1)	295 × 115 and 539 × 94	Rod-shaped	BBB	[157]
	Lipid	72–122	Star-shaped		[162]
	Gold nanorods functionalized with 4-mercaptophenol	40 × 12	Rod-shaped		[159]
	Polystyrene rods	400 × 200	Rod-shaped		[243]
	11-mercaptoundecanoic acid coated Au cores	50	Star-shaped		[161]
	Polystyrene rods	301 × 120	Rod-shaped		[156]
	Au rods and stars coated with carboxy-PEG thiol ligand	60 × 30 and 55	Rod/star-shaped		[160]
	TiO_2_ rods	40 × 21	Rod-shaped		[244]
	PEG-PEI copolymer coated mesoporous silica rods	300 × 100	Rod-shaped		[158]
Magnetic-field-enhanced permeation	Iron oxide core coated by gold and conjugated with PEG	38 and 77	Spherical	BBB	[168]
	Discrete model for the magnetic NPs	10 and 100		BBB	[169]
	EDT-coated iron oxide cores loaded with DOX	72–79		BBB	[163]
	PEI-PEG-coated iron oxide cores loaded with Salinomycin	70–96		BBB	[170]
	Carboxymethyl cellulose coated iron oxide cores	12–16		BBB	[164]
	Au-coated magnetite cores loaded with DOX	100		Blood–spine barrier	[165]
	Silica-coated magnetite core loaded with cell-penetrating peptide Tat	84–91		BBB	[166]
	Aminosilane-coated and EDT-coated iron oxide cores	25 and 29		BBB	[167]
The effect of focused ultrasound	Red-blood-cell membrane cloaked liposomes with perfluorocarbon	144	Spherical	BBB	[174]
	Dual fluorophore-labeled core-crosslinked polymeric micelles	65			[177]
	Lipid–polymer hybrid NPs loaded with CRISPR/Cas9 plasmids and modified with the cRGD peptide	135–235			[172]
	Quercetin-modified sulfur NPs	38–68			[175]
	PEGylated Au cores	3, 15, 120			[176]
	Gd chelates: Gd-DOTA (Dotarem^®^), Gd-DO3A-butrol (Gadovist^®^), Gd-BOPTA (MultiHance^®^)	1–2			[178]
	Polyacrylic-acid-coated Au cores with uptake peptide conjugated with Cis	7			[179]
	PEGylated Au cores	10 and 50			[173]
Electroporation	Gd-DOTA (Dotarem^®^)	~1	n.a.	BBB	[183]
	Dye-stabilized Sorafenib NPs	83–89	Spherical	BTB	[180]
	Gadopentetate dimeglumine	~1	n.a.	BBB	[185]
	Fluorescein	<1	n.a.	BBB	[182]
	Fluorescein Isothiocyanate-Dextran	~1	n.a.	BBB	[181]
	Gd-DOTA (Dotarem^®^)	~1	n.a.	BBB	[184]
	Gd-DOTA (Dotarem^®^)	~1	n.a.	BBB	[186]
The effect of electromagnetic radiation	Iron oxide cores coated with PLGA and PEG	5	Spherical	BBB	[189]
	ICG and carboxymethyl chitosan modified size-tunable Au cores	41–65	Spherical	BTB	[196]
	mPEG-stabilized Au cores	50	Spherical	BBB	[192]
	Bradykinin conjugated self-assembled aggregation-induced-emission NPs	100	Spherical	BTB	[193]
	Au NPs and rods	n.a.	Spherical and rod-shaped	BBB	[194]
	PEG-stabilized Au cores conjugated to RGD-peptide	5–10	Spherical	BTB	[245]
	Thermoresponsive lipid NPs loaded with paclitaxel	68–238	Spherical	BBB	[246]
	Poly(maleic acid-co-olefin)-coated magnetite cores	3–18	Spherical	BBB	[187]
	Cross-linked nanoassemblies loaded with superparamagnetic iron oxide NPs	22–28	n.a.	BBB	[188]
	PEI-coated/poly-(γ-glutamic acid)/PLGA NPs loaded with saquinavir	210–450	Spherical	BBB	[190]
	Tetramethylrhodamine-conjugated magnetic oxide NPs; adenovirus	n.a.; ~100	Spherical	BBB	[195]
	SLNs, PBCA, and MMA-SPM NPs loaded with saquinavir	135; 92; 8	Spherical	BBB	[191]
Magnetic field and ultrasound combined effect	Gene-loaded PEI-modified magnetic mesoporous silica NPs	59–105	Spherical	BTB	[197]
Photoacoustic cavitation	Den-RGD/CGS/Cy5.5 NPs	10	Spherical	BBB	[198]
Intracellular tension modulation	Protein NPs	10–100	Spherical	BBB	[93]

Abbreviations: NPs—nanoparticles; BBB—blood–brain barrier; BTBB—blood–tumor brain barrier; POMs—polyoxometalate; BAB—blood–air barrier; mPEG-PLGA—monomethoxy (polyethylene glycol) d,l-lactic-co-glycolic acid; SLNs—solid lipid nanoparticles; ZP—Zeta potential; PLGA—poly(lactic-co-glycolic acid); BPB—blood–placenta barrier; PEG—polyethylene glycol; PEG-PEI—poly(ethylene imine); EDT—trimethoxysilylpropyl-ethylenediamine triacetic acid; DOX—doxorubicin; ICG—indocyanine green; PBCA—polybutylcyanoacrylate; MMA-SPM—methylmethacrylate-sulfopropylmethacrylate; Den-RGD/CGS/Cy5.5—den-cyclo (Arg-Gly-Asp-d-Tyr-Lys)(RGD)/4-[2-[[6-Amino-9-(*N*-ethyl-β-d-ribofuranuronamidosyl)-9*H*-purin-2-yl]amino]ethyl]benzenepropanoic acid hydrochloride (CGS)/Cy5.5.

## 5. Conclusions

The development of novel biological and physicochemical methods to increase the passing of nanocarriers across the blood–tissue barriers in order to increase their intratumoral accumulation represents one of the trends in translational oncology. To date, the reported techniques in preclinical studies have demonstrated significant efficacy in increasing the penetrative properties of the particles. Presumably, a combination of several methods (e.g., decoration of nanoparticle surface by CPPs and application of either chemical agents (inhibitors of tight junctions or efflux pump) or physical methods) could further potentiate the targeting properties of the particles, but this should be tested in further translational studies.

## Figures and Tables

**Figure 1 nanomaterials-13-01140-f001:**
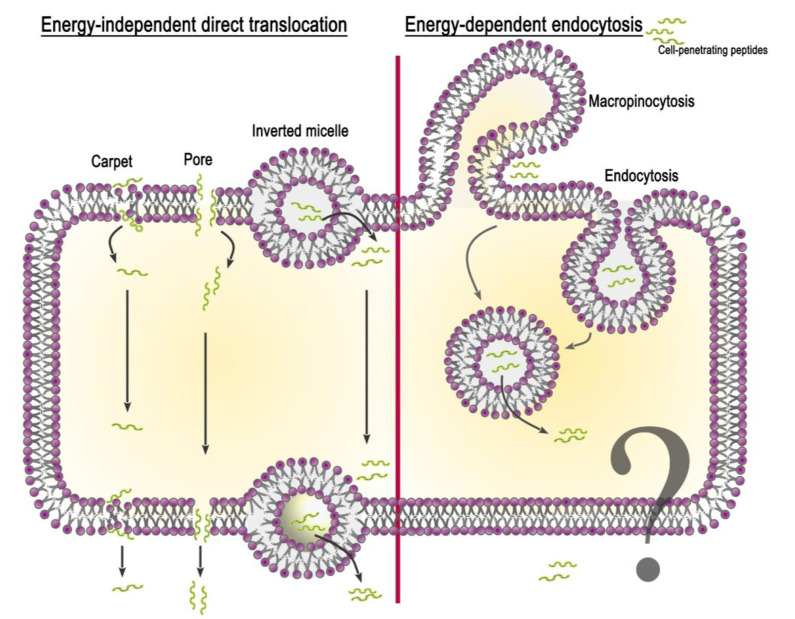
Schematic representation of proposed mechanisms for CPP internalization.

**Figure 2 nanomaterials-13-01140-f002:**
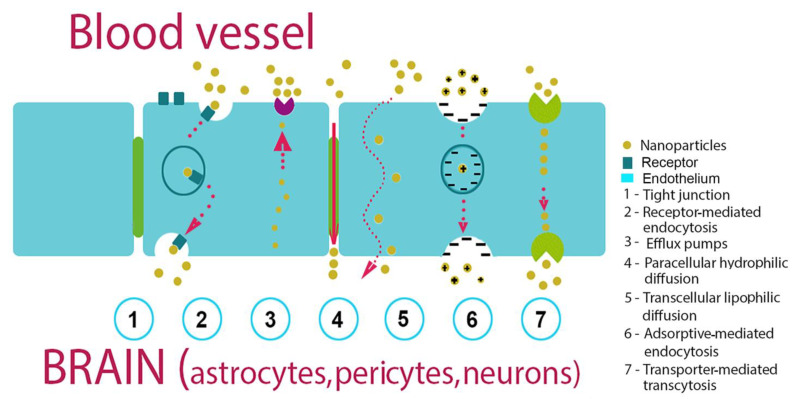
Mechanisms of NPs penetration via the histohematic barriers (blood–brain barrier (BBB) used as an example).

**Figure 3 nanomaterials-13-01140-f003:**
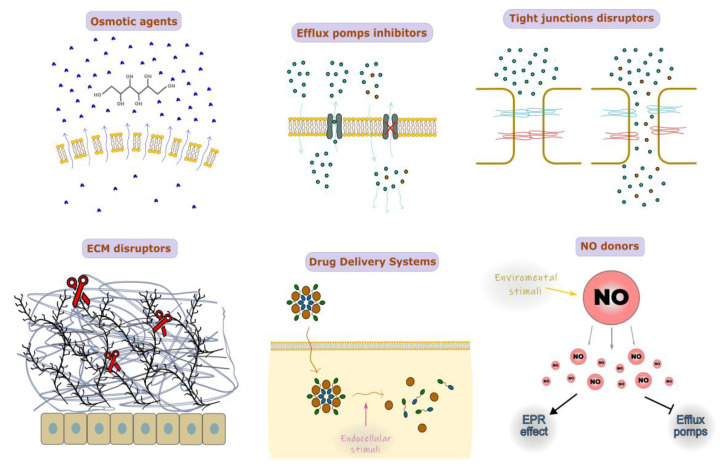
Chemical modification of the histohematic barriers permeability.

**Figure 4 nanomaterials-13-01140-f004:**
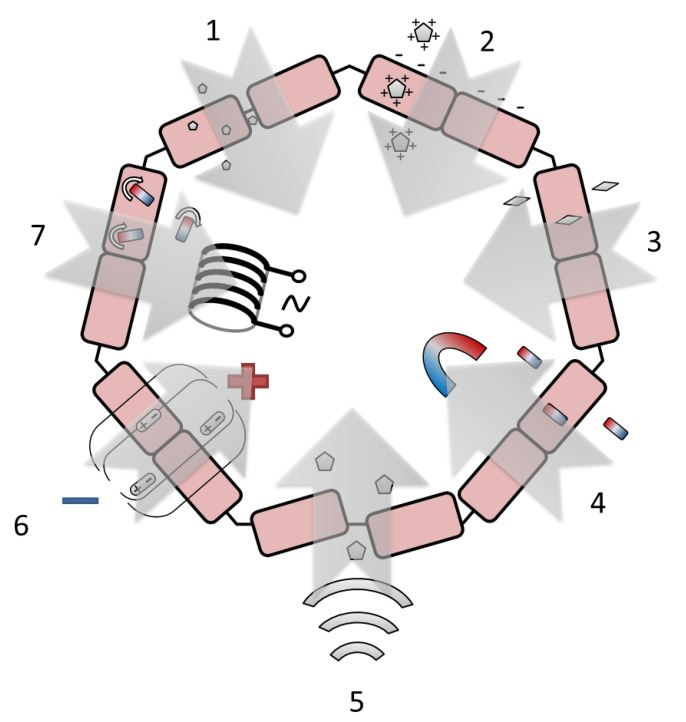
The most frequently proposed approaches to enhance the histohematic barrier permeability: 1—NP size reduction, 2—NP surface charge modulation, “+” and “–” conditionally designate various surface charge, 3—NP shape modulation, 4—magnetic-field-enhanced permeation, 5—the effect of focused ultrasound, 6—electroporation, “+” and “–” designate electrical poles, 7—the effect of electromagnetic radiation, rounded arrows indicate the rotation of NPs under the action of an alternating field created by an electromagnetic coil.

## Data Availability

MDPI Research Data Policies.

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
