# Peer review of "Passing of Nanocarriers across the Histohematic Barriers: Current Approaches for Tumor Theranostics"

_nanomaterials, 2023, doi:10.3390/nano13071140_

Round 1

Reviewer 1 Report

Overall a well written review manuscript on an interesting topic. However authors should better proofread the manuscript co ensure the consistency and clarity.

For example usage of et cetera within the text is not consistant. at some instances it is written with a fullstop and sometimes without.

Abbreviations: some are used but the emaning is not explained an can be misinterpreted by some readers (i.e. BBTB, BPB and ABB in Tbales 1, 3 and 4, or GMB and BDD in Table 2). Explanation consistency: different styles of explanation. Look at legend of Tables 2 and 3.

BBTB used in the tables is should stand for Blood-Tumor Brain Barrier. Should it be then abbreviated as BTBB?

Or abbreviation for Cell Penetrating Peptides (CPP) is misspelled from time to time (instead of CPP is used CCP).

Figure 2 is too small, thus hard to decipher the meaning of it.

Author Response

We would like to thank the reviewer for the provided comments. We have carefully revised the manuscript according to the comments.

QUESTION 1: Abbreviations: some are used but the emaning is not explained an can be misinterpreted by some readers (i.e. BBTB, BPB and ABB in Tbales 1, 3 and 4, or GMB and BDD in Table 2). Explanation consistency: different styles of explanation. Look at legend of Tables 2 and 3.

ANSWER 1: We have revised the description of the tables and corrected the abbreviations.

QUESTION 2: BBTB used in the tables is should stand for Blood-Tumor Brain Barrier. Should it be then abbreviated as BTBB?

ANSWER 2: We have corrected to BTBB throughout the manuscript.

QUESTION 3: Or abbreviation for Cell Penetrating Peptides (CPP) is misspelled from time to time (instead of CPP is used CCP).

ASNWER 3: This was corrected.

QUESTION 4: Figure 2 is too small, thus hard to decipher the meaning of it.

ANSWER 4: We have revised the Figure 2.

Reviewer 2 Report

This is a review of the manuscript entitled "Passing of nanocarriers across the histo-hematic barriers: current approaches for tumor theranostics" submitted by Kamil Gareev.

The review is rich and meaningful, focusing on approaches to surface functionalization with cell-permeable peptides and increasing the permeability of blood-tissue barriers (BTB) in order to break through BTB in nanoparticle-based tumor therapy and diagnosis, leading to a system that is applicable in clinical setting. This manuscript is well described.  Therefore, I recommend this manuscript for publication after minor revision:

1) Although the authors notes ”Nanoparticle-based systems in recent decades have been employed for the tumor diagnostics and therapy (i.e., theranostics)” and gives its citation as reference 1 at P1 line 36-37, the cited reference is about cancer immunotherapy, which is partially out of sync with this text. It is better to add or replace the reference 1 with a paper that takes a broader view of nanoparticles as applied to tumor diagnosis and therapy.

2) At P7 line 216-218, authors described “It is assumed that TAT passes through the cell membrane by direct penetration, but when peptide is conjugated with a cargo, the mechanism of energy-dependent endocytosis is activated”. It is appropriate to add a condition such as "at low concentrations" to indicate that TAT without cargo is directly permeable, and if the condition is ambiguous, it gives the reader the impression that it is inconsistent with the description on P4 lines 174-176. It is reported that TAT not bound to Cargo is predominantly taken up by direct permeation at low concentrations (<10uM) and by clathrin-mediated endocytosis at higher concentrations. (Falk Duchardt et al., Traffic 2007, 8, 848–866)

It would be appropriate to add this paper as a citation in the text.

3) At P11 line 424, it is difficult to understand what the numbers at the bottom of Figure 2 indicate. The position of the numbers at the bottom of Figure 2 need to be improved.

Author Response

We would like to thank the reviewer for the provided comments. We have carefully revised the manuscript according to the provided comments.

QUESTION 1: Although the authors notes ”Nanoparticle-based systems in recent decades have been employed for the tumor diagnostics and therapy (i.e., theranostics)” and gives its citation as reference 1 at P1 line 36-37, the cited reference is about cancer immunotherapy, which is partially out of sync with this text. It is better to add or replace the reference 1 with a paper that takes a broader view of nanoparticles as applied to tumor diagnosis and therapy.

ANSWER 1: We have substituted the reference.

QUESTION 2: At P7 line 216-218, authors described “It is assumed that TAT passes through the cell membrane by direct penetration, but when peptide is conjugated with a cargo, the mechanism of energy-dependent endocytosis is activated”. It is appropriate to add a condition such as "at low concentrations" to indicate that TAT without cargo is directly permeable, and if the condition is ambiguous, it gives the reader the impression that it is inconsistent with the description on P4 lines 174-176. It is reported that TAT not bound to Cargo is predominantly taken up by direct permeation at low concentrations (<10uM) and by clathrin-mediated endocytosis at higher concentrations. (Falk Duchardt et al., Traffic 2007, 8, 848–866)

It would be appropriate to add this paper as a citation in the text.

ANSWER 3: We have corrected this paragraph and added the suggested manuscript into the references [45].

QUESTION 3: At P11 line 424, it is difficult to understand what the numbers at the bottom of Figure 2 indicate. The position of the numbers at the bottom of Figure 2 need to be improved.

ANSWER 3: We have revised the Figure 2.